



# Topological Analysis in Monte Carlo Simulation for Uncertainty Estimation

Evren Pakyuz-Charrier[1,3], Mark Jessell[1], Jérémie Giraud[1], Mark Lindsay[1], Vitaliy Ogarko[2],

[1]: Centre for Exploration Targeting, The University of Western Australia, 35 Stirling Hwy, Crawley WA 6009 Australia;

[2]: International Centre for Radio Astronomy Research, The University of Western Australia, 35 Stirling Hwy, Crawley WA 6009 Australia;

[3]: Intrepid Geophysics, 3 Male Street, Brighton VIC 3186 Australia;

Corresponding Author: E. Pakyuz-Charrier evrenpakyuzcharrier@gmail.com)

**Abstract.** This paper proposes and demonstrates improvements for the Monte Carlo simulation for Uncertainty Estimation (MCUE) method. MCUE is a type of Bayesian Monte Carlo aimed at input data uncertainty propagation in implicit 3D geological modeling. In the Monte Carlo process, a series of statistically plausible models are built from the input data set which uncertainty is to be propagated to a final probabilistic geological model (PGM) or uncertainty index model (UIM).

Significant differences in terms of topology are observed in the plausible model suite that is generated as an intermediary step in MCUE. These differences are interpreted as analogous to population heterogeneity. The source of this heterogeneity is traced to be the non-linear relationship between plausible datasets' variability and plausible model's variability. Non-linearity is shown to arise from the effect of the geometrical ruleset on model building which transforms lithological continuous interfaces into discontinuous piecewise ones. Plausible model heterogeneity induces geological incompatibility and challenges the underlying assumption of homogeneity which global uncertainty estimates rely on. To address this issue, a method for topological analysis applied to the plausible model suite in MCUE is introduced. Boolean topological signatures recording lithological units' adjacency are used as n-dimensional points to be considered individually or clustered using the Density-Based Spatial Clustering of Applications with Noise (DBSCAN) algorithm. The proposed method is tested on two challenging synthetic examples with varying levels of confidence in the structural input data.

Results indicate that topological signatures constitute a powerful discriminant to address plausible model heterogeneity. Basic topological signatures appear to be a reliable indicator of the structural behavior of the plausible models and provide useful geological insights. Moreover, ignoring heterogeneity was found to be detrimental to the accuracy and relevance of the PGMs and UIMs.

KEYWORDS: uncertainty propagation, 3D geological modeling, Monte Carlo, topology, DBSCAN, clustering

**Highlights**

- Monte Carlo uncertainty estimation (MCUE) methods often produce geologically incompatible plausible models
- Geologically incompatible models can be differentiated using topological signatures
- Geologically consistent probabilistic geological models (PGM) may be obtained through topological signatures clustering



## Introduction

Input data uncertainty propagation in is an essential part of risk aware 3D geological modeling (Schweizer et al., 2017;Wang et al., 2016;Nearing et al., 2016;Aguilar et al., 2018;Mery et al., 2017;Dang et al., 2017;Lark et al., 2013). Accurate quantification of geometrical uncertainty is indeed key to determine the degree of confidence one can put into a model. How reliable a 3D geological
model is and how this reliability varies in space are indispensable data to seek improvement of said model. Monte Carlo based uncertainty estimation (MCUE) algorithms have recently been proposed to tackle this issue (de la Varga and Wellmann, 2016;Pakyuz Charrier et al., 2018;Pakyuz-Charrier et al., 2018). MCUE methods (Figure 1) aim to propagate the measurement uncertainty of structural input data (interface points, foliations, fold axes) through implicit 3D geological modeling engines to produce probabilistic geological models (PGM) and uncertainty index models (UIM). To do so each structural input data is replaced
by a probability distribution thought to best represent its measurement uncertainty called a disturbance distribution (Pakyuz-Charrier et al., 2017b). Disturbance distributions are then sampled using Markov-Chain Monte-Carlo or random methods to generate alternative statistically plausible datasets. Plausible datasets can then used to build a suite of plausible 3D geological models which may be merged into PGMs or UIMs. Recent works (Thiele et al., 2016a;Thiele et al., 2016b) have demonstrated that the plausible 3D geological models' suite may display great geometrical variability to the point of making some plausible models conceptually
incompatible with one another. Plausible models' incompatibility is damaging to the relevance of MCUE because the PGMs and UIMs implicitly assume plausible model homogeneity.

In this paper, the standard MCUE procedure is described, the source of plausible models' incompatibility is discussed, and a topological analysis method is proposed to address the issue and improve the relevance of PGMs and UIMs to real world problems. The method relies on the extraction of adjacency matrices for each plausible model. Adjacency matrices qualify which geological
units are in contact using Boolean logic. These matrices are then converted to binary signals called topological signatures that are then clustered using DBSCAN. The goal is to provide MCUE practitioners with a procedure to ensure that PGMs and UIMs are made of topologically consistent plausible models. Lastly, the method is tried and tested on two synthetic case studies to demonstrate its applicability.

## MCUE method

Monte Carlo simulation for Uncertainty Estimation (MCUE) is an uncertainty propagation method focusing on input structural data (interface points, foliations, fold axes, drillhole data). It is usually applied to implicit 3D geological modeling (Giraud et al., 2017). Note that MCUE shares aspects with the Generalized Likelihood Uncertainty Estimation (Beven and Binley, 1992), a special implementation of a Bayesian Monte Carlo approach (Camacho et al., 2015). MCUE aims to provide probabilistic models and estimate model uncertainty by producing a range of alternate plausible 3D geological models and performing comparative analysis
on them (Pakyuz-Charrier et al., 2017a;Wellmann, 2013;Lindsay et al., 2013). 3D geological model suites are built from a series of plausible datasets that are generated through input data perturbation (Figure 1), which is a process in which alternative input datasets are stochastically generated from the original data inputs by sampling from probability distribution functions known as disturbance distributions (Pakyuz-Charrier et al., 2017b).

### Disturbance distribution parameterization

Disturbance distributions are probability distribution functions that are used to generate plausible datasets in MCUE. They are designed to simulate the effect of the inherent uncertainty of each observation separately. In principle, an individual disturbance distribution is associated to each observation (Figure1, preprocessing). Disturbance distributions are expected to be chosen and parameterized based on thorough metrological analysis of the original dataset, since disturbance distributions are expected to aggregate as many sources of input data uncertainty as possible. These sources of uncertainty relate to measurement error, rounding
error, user error, local variability, mis-calibration or projection issues (Bardossy and Fodor, 2001). Generally, Gaussian-like distributions make for appropriate disturbance distributions (Pakyuz-Charrier et al., 2017b). Disturbance distribution selection and



parameterization is a complex topic and is outside the scope of this paper. However, practitioners may seek guidance from recent practical metrological work on foliations (Novakova and Pavlis, 2017;Stigsson, 2016;Cawood et al., 2017) and more theoretical work on disturbance distribution selection/parameterization for MCUE (de la Varga and Wellmann, 2016;Pakyuz-Charrier et al., 2017b).

**Plausible datasets generation**

Plausible datasets are obtained by sampling from the numerous disturbance distributions that have been defined for each input observation. The sampling step is often referred to as the "perturbation" of the input data (Cherpeau et al., 2010). Sampling is usually made independently as errors do not show any spatial dependency, because measurements are physically independent (Pakyuz-Charrier et al., 2017b). Nevertheless, spatial correlation of errors can be observed. Indeed, observing heteroscedasticity in the
original dataset would imply that some level of error spatial correlation is possible. This is especially true for cyclical datasets such as foliations in folding scenarios. The sampling step may be followed by a range statistical checks to ensure stationarity, reject outlying datasets, examine likelihood or perform variographic analysis.

**Plausible models building**

Plausible dataset generation is an important part of the MCUE method because it heavily predetermines its outcomes. However,
plausible datasets are only as relevant as the plausible model they correspond to. MCUE is then largely dependent on the particulars of the chosen modeling engine (Figure1, Building). Any modeling engine relies on the conceptualization of the phenomenon it is supposed to model. Conceptualization relies mainly on abstraction and simplification to make the modeling problem accessible to our minds and technology. Therefore, any workflow or method that relies on a modeling engine subsequently relies on these abstractions and simplifications which, by definition, are incomplete and uncertain. Consequently, MCUE is sensitive to this kind
of "conceptual uncertainty" and care should be taken when selecting or parameterizing the modeling engine. Given that MCUE's main aim is to propagate input uncertainty through the modeling engine to the final model, several indispensable properties of the modeling engine may be identified (i) the ability to estimate and propagate its own uncertainty (ii) may handle multiple plausible datasets without having to be reconfigured manually (iii) does not rely extensively on expert input. These properties are generally met by implicit modeling engines (Chilès et al., 2004;Aug et al., 2005;Calcagno et al., 2008;Chilès and Delfiner, 2009) by the virtue
of them being reliant on potential field interpolation to estimate the geological surfaces from the input structural data. The interpolator is normally parameterized using variographic analysis and a geometrical ruleset to solve geometrical ambiguities (Jessell, 2001). The geometrical ruleset consists of a series of geometrical constraints such as the intersection priority of faults and geological units that are used to determine which interface stops on which. Conceptually, the geometrical ruleset enforces the age relationships between the faults and/or geological units in the model. In this paper, the modeling engine is the GeoModeller software
which uses a stochastic cokriging interpolator and constrains surfaces using a predefined stratigraphic pile and fault relationship matrices as geometrical ruleset (Guillen et al., 2008;Calcagno et al., 2008).

**Comparative analysis**

In implicit 3D geological modeling, a model is essentially a set of spatial functions that describe the geometry of stratigraphic and intrusive interfaces and fault planes. In this form, it is difficult to apply common comparative analysis methods. Therefore, plausible
models are either discretized to 3D grids (voxets), flattened to triangulated surfaces or shrunk to triple lines (Figure1, Postprocessing). Note that in all three cases, these operations are further simplifications of the models and add more uncertainty to the final outcome. Each of these transformations allow for different comparative analyses to be run (i) voxets are used to build probabilistic geological models (PGM) and uncertainty index models (UIM) such as entropy or stratigraphic variability (Wellmann and Regenauer-Lieb, 2012;Lindsay et al., 2012) (ii) the shape of triangulated surfaces may be used to estimate the variability of
curvature (Lindsay et al., 2013) (iii) triple lines may be used to analyze space partitioning in a more compact way than that of the previous two forms. Furthermore, the results of these analyses can be fed to external validation systems to reduce geological



uncertainty and improve understanding of the modelled volume. Examples of external validation systems include geophysical inversion (Giraud et al., 2019), concurrent geophysical forward modeling (Bijani et al., 2017;Lipari et al., 2017), or groundtruthing. Lastly, the results obtained from the external validation systems may be reutilized by MCUE to further refine the models.

**Plausible model heterogeneity**

As stated in the previous section, comparative analysis in MCUE aims to study the variability of the plausible models and extract meaning from them. To this end, plausible models are transformed to a more manageable form that fits our analysis tools (Figure 1). The most common comparative analysis tools used in MCUE are UIM such as Information Entropy and stratigraphic variability. These indexes are computed from a relative frequency voxet that is obtained by merging the voxets from all of the plausible models together. The underlying assumption is that the plausible models constitute a unimodal population and may be analyzed as such.

The UIM used in MCUE are scalar proxies for categorical uncertainty and one of the critical conditions for a single scalar to be representative of the uncertainty of a variable is that it has to be distributed unimodally. To assume unimodality is risky because it restrains the relevance of the UIM to homogenous populations only. In the case of a heterogeneous population or a mixture of populations, this procedure will fail to represent accurately the behavior of the variable in the same way a bimodal distribution cannot be fully described by its mean and variance (Figure 2). In the case of MCUE, perturbation is usually performed using

unimodal gaussian disturbance distributions (Pakyuz-Charrier et al., 2018;Pakyuz Charrier et al., 2018) and at first sight it may seem that model building should result in a homogenous population of plausible models. However, it has been demonstrated on simple synthetic cases that plausible models with strikingly different structural geological features may arise from perturbing the same original dataset (Thiele et al., 2016a;Thiele et al., 2016b) using unimodal disturbance distributions (Figure 3). These differences indicate that standard perturbation may lead to plausible model heterogeneity. This effect stems from the fact that the relationship

between the variability of the plausible datasets and that of their corresponding plausible models is non-linear (Figure 3, Figure 4). The non-linearity of the plausible model suites can be explained by the interactions between the interpolator and the geometrical ruleset. The interpolators used in implicit 3D geological modeling are linear (Kriging, RBF) and it is the geometrical ruleset that introduces non-linearity by adding a discrete component to model realization. For example, a plausible model suite may display the same fault in various scenarios (normal, reverse, decollement) or open/close potential traps for fluids (Figure 4). In the latter example

(Figure 4), non-linearity is observed because of the geometrical ruleset that gives intersection priority to the top impermeable unit (green) over the lower units. If not for this ruleset, interfaces would vary linearly, and no unit would stop on any other unit. Consequently (i) very small changes in a plausible dataset may induce large changes in the subsequent plausible model (ii) plausible dataset variography is not a reliable indicator of plausible model homogeneity. Therefore, standard statistical filters applied to plausible datasets are unlikely to prevent or warn of potential plausible model heterogeneity. Special sampling methods are such as

Gibbs sampling may decrease model variability by forcing internal spatial correlation in plausible datasets (Wang et al., 2016) although, as stated above this is not guaranteed. Moreover, these methods work best if errors are spatially dependent. This is normally not the case for sparse geological structural measurements taken individually. Actually, there is no logical reason to consider that the measurement errors related to, for example, two foliations measured with a compass in different areas are dependent on one another. Note that spatial correlation of errors remains possible when the original dataset is heteroscedastic and measurements

themselves are spatially correlated.

**Plausible model topological analysis**

As ignoring plausible models suites' heterogeneity may lead to an unknown amount of knowledge degradation, the need to distinguish and classify plausible models that correspond to different geological scenarios becomes apparent. By doing so, it becomes possible to design a scenario-based comparative analysis step in MCUE. In principle this approach has multiple advantages,

a geological scenario-based procedure can be expected to (i) allow rejection of physically absurd models (ii) reduce uncertainty on



a per-scenario basis (iii) enable targeted improvement of the model. A common way to distinguish groups or trends in complex dataset is via the use of clustering algorithms or machine learning. In MCUE, clustering is preferable because machine learning relies on training and validation datasets to function properly. Unfortunately, MCUE does not provide a reliable way to determine the adequacy of a plausible model training dataset for machine learning beforehand. In contrast, and given a certain number of

assumptions, clustering algorithms are expected to work with the raw data. In this paper, the Density-Based Spatial Clustering of Applications with Noise (DBSCAN) (Ester et al., 1996) was selected for its simplicity, speed, robustness and overall reliability (Chakraborty et al., 2014;Schubert et al., 2017). However, all clustering algorithms require a relevant discriminatory variable to build clusters efficiently. In this instance, the discriminatory variable has to be logically linked with plausible models' heterogeneity to allow the clustering algorithm to differentiate geological scenarios. A potential candidate that meets this criterion is lithological

topology which was recently demonstrated to be an efficient tool to recognize highly discriminating features from plausible models in MCUE (Wellmann et al., 2015;Thiele et al., 2016a). As stated in the previous sections, the non-linearity and non-uniqueness in 3D geological modeling is the cause of plausible model heterogeneity. In addition, non-linearity and non-uniqueness result from the topological constraints imposed by the geometrical ruleset. Therefore, the geometrical ruleset is at least partially responsible for the plausible models' heterogeneity. It is then reasonable to assume that the topology of the plausible models can be used as a

discriminatory variable to combat model heterogeneity.

**Lithological topology**

Topology describes the properties of special mathematical spaces that are unaltered under continuous deformation (Crossley, 2006). 3D geological modeling mostly concerns itself with the topic of geospatial topology that focusses on spatial relationships such as adjacency, overlap or separation of geometrical objects such as points, lines, polygons and polyhedrons (Thiele et al., 2016a).

Formally, eight binary spatial relationships are possible between 2D objects (Egenhofer, 1989). There is a total of sixty-one cross-dimensional binary spatial relationships between 0, 1, 2 and 3D objects (Zlatanova, 2000). Essentially, the use of topological relationships to characterize 3D geological models allows a compact expression of a sub-set of their geometry (Burns, 1988). Combined with the knowledge of the intrinsic physical properties of the rock types that compose geological units, these relationships constrain the downstream predictions resulting from 3D geological models in terms of physical processes such as fluid, heat flow

and electrical flow as well as mechanical stresses. The most common relationships between 3D objects encountered in 3D geological models are adjacency and separation of lithological units. In their simplest form, these relationships can be expressed using an adjacency matrix. Each element of the adjacency matrix is a boolean where 0 encodes separation and 1 encodes adjacency (Figure 5). However, an adjacency matrix contains both redundant and irrelevant information. Indeed, the adjacency matrix $A$ of a model $M$ comprised of $n$ geological units is symmetric and hollow. A is then of size $n^2$ with its diagonal comprised solely of 1 while both

sides are transpose of one another, it is then useful to half-vectorize $A$ and remove unit elements from the diagonal following the triangular number sequence. For example, the $4 \times 4$ adjacency matrix

(1)

$$A = \begin{bmatrix} 1 & 1 & 0 & 1 \\ 1 & 1 & 1 & 1 \\ 0 & 1 & 1 & 0 \\ 1 & 1 & 0 & 1 \end{bmatrix},$$

is half vectorized

(2)

$$\text{vech}(A)^T = \begin{bmatrix} 1 & 1 & 0 & 1 & 1 & 1 & 1 & 1 & 0 & 1 \end{bmatrix},$$

Note that vech($A$) is of size $\frac{n^2+n}{2}$ and contains all the necessary information to fully describe the adjacency of lithological units in a 3D geological model with $n$ distinct lithological units. vech($A$) can be also considered as a $\frac{n^2+n}{2}$ bit binary sequence called a basic topological signature. The total number of possible topological signatures is $2^{\frac{n^2+n}{2}}$. However, it is unlikely that all possible





signatures are present in the plausible model suite given that the geometrical ruleset constrain their topology. Consequently, the issue of the representativity of the plausible model suite in terms of the variability of its topological signatures comes into question.. At a minimum, the variability of topological signatures should be qualitatively representative of the plausible model space to allow clustering algorithm to delineate the right number of clusters. Cumulative observed topological signatures graphs are a practical and

efficient way to determine the topological representativity of the plausible model suite in real time (Thiele et al., 2016b). As the modeling engine produces new plausible models, these graphs plot the number of distinct topological signatures observed versus the number of plausible models generated so far. When the number of distinct topological signatures observed reaches a plateau, it is safe to consider that most topologies have been observed and qualitative topological stationarity may then be assumed reasonably (Figure 6). Note that clustering the topological signatures of the plausible model suite implies that quantitative topological

stationarity is not required. That is, distinct topological signatures need not to be in exact proportions relative to each other given that the clustering algorithm is expected to have them separated.

**Topological clustering using DBSCAN**

The Density-based spatial clustering of applications with noise (DBSCAN) is a point density reliant flat data clustering algorithm (Schubert et al., 2017;Ester et al., 1996). DBSCAN is based on the notion on the reachability of border points from core points

(Figure 7). The algorithm only needs two parameters (i) the minimum number of points $P_{min}$ that are required to form a cluster and (ii) the maximum distance $\varepsilon$ allowed for two points to still be considered to be neighbors. On this basis the algorithm builds a distance matrix between all points and uses that matrix to determine the neighbors of each points based on $\varepsilon$. Each points that has at least $P_{min}$ neighbors is a core point that forms a cluster seed to which all directly reachable points are attached. In order to build the distance matrix, DBSCAN requires each point to be characterized by a metric variable. Therefore, the variable would allow

distances to be computed using regular norms such as the Euclidean distance. However, topological signatures form a series of Boolean variables that cannot provide appropriate measures for they are not additive. An alternative is to consider the whole topological signatures as a binary word and use the Hamming distance (Hamming, 1950) as the metric. The Hamming distance counts the number of individual bit switches required to match two binary words of equal lengths, effectively quantifying their disagreement. Implementation wise, a simple XOR over two topological signatures gives the Hamming distance that separates them.

As a special case, when $\varepsilon = 0$ and $P_{min} = 1$, DBSCAN will distinguish every distinct topological signature into a separate cluster and the size of each cluster will count their occurrences.

**Post-clustering analysis**

Once the plausible model suite has been segregated into clusters based on their topology, a range of statistical methods may be applied to the results to (i) evaluate the quality and relevance of the clusters (ii) determine data leverage in relation to the clusters

(iii) perform standard MCUE comparative analysis on the clusters (iv) feed the clusters to an external rejection system. Cluster quality may be evaluated by computing the internal information Entropy matrix $E$ for each cluster

(3)

$$E_i^j = -\sum_{k=1}^{c} A(k)_i^j \log\big(k A(k)_i^j\big),$$

where $A(k)$ is the $k^{th}$ adjacency matrix of the cluster, $c$ is the cardinality of the cluster and $i, j$ are standard matrix indexes. For a

given cluster, $E$ informs the user about the internal variability of the binary topological relationships between each lithological couple. Most entries are expected to be null, indicating no variations, while non-null entries indicate topological "switches" inside the cluster itself. That is, E highlights topological changes that the clustering algorithm considered not to be significant enough to warrant a split in the cluster. Importantly, this is directly translatable into geological insights: "these two models are different because in only one of them is the sandstone unit found adjacent to the shale unit". Naturally, (3 may be applied to the whole suite of

plausible models' adjacency matrices as a practical reference to compare individual clusters' internal information Entropy matrices



to a global information Entropy matrix. Standard MCUE comparative analysis tools may be applied to the individual clusters concurrently to, for example, obtain per-cluster/scenario uncertainty indexes and sub-PGMs. Given that common MCUE UIM are sums of matching positive elements, per-cluster UIM voxels are guaranteed to yield equal or lower values compared to their global equivalent. Moreover, per-cluster UIM are expected to be better structured as a common effect of all clustering algorithms is to reduces intraclass variance. Clustered plausible models may be traced back to their plausible input datasets (structural measurements) to conduct cluster leverage analysis. The aim of cluster leverage analysis is to determine which parts of the datasets are responsible for the topological switches that induce the formation of new clusters. A straightforward way to achieve this aim is to compute a central statistic for every individual input in every cluster's plausible datasets

(4)

$$\bar{\mathbf{u}} = [\bar{d}_{l=1} \quad \dots \quad \bar{d}_t],$$

where $\bar{\mathbf{u}}$ is the vector of central, $\bar{d}_l$ is the central statistic for the plausible input observation $l$ and t is the cardinality of the input data. The next step is to compare every matching individual input data central statistic between all cluster pairs

(5)

$$\Delta\bar{\mathbf{u}}(a,b) = (\bar{\mathbf{u}}_a - \bar{\mathbf{u}}_b) \circ (\bar{\mathbf{u}}_a - \bar{\mathbf{u}}_b).$$

Where $a, b$ identifies a cluster pair and ∘ stands for the Hadamard product. The results of this procedure should be ranked to find the highest leverage plausible input data differences between clusters.

**Synthetic case study**

To serve as proof of concept, the plausible models clustering procedure that is proposed in the previous section is tested on a synthetic case of medium complexity called CarloTopo. The aim is to assess how plausible model clustering may improve the accuracy, practicability and tractability of MCUE in a comprehensible yet relevant environment. The procedure follows standard MCUE (Figure 1) with topological clustering being added to the last step of comparative analysis. Results are expressed in three complementary modes, (i) differences between topological clusters are visualized using information Entropy as a proxy for uncertainty estimation; (ii) intra-cluster variability is assessed using internal Entropy matrices; (iii) the initial and individual plausible models are characterized by their topological signatures and lithological cross-sections.

**Model description and MCUE parameters**

The CarloTopo 3D geological model features 8 lithological units distributed into 5 series and 2 faults (Figure 8). All of the 25 foliations and 46 interface (Table 1) points for all units and faults are placed onto a single N-S vertical median cross-section. This design decision was made to ensure that the cross sections discussed in the subsequent sections are representative of the models. CarloTopo simulates a normally faulted basin placed on top of a mafic formation that sits on an erosional surface. Below the erosional surface is a metamorphic folded series comprised of 3 individual formations. The metamorphic series rests onto the basement and both are intruded by a pluton. The geometries for each unit were designed to manifest as many common geological features as possible without compromising its relevance for practical issues such as mining/oil & gas exploration. More specifically, several potential traps for sedimentary-hosted deposits were included in the original model along with a, network of faults that serve as theoretical channels or barriers (Figure 9). The case study was split into two separate MCUE experiments with different disturbance distribution parameterization with over a thousand perturbations each. The first run aims to simulate a high input data confidence scenario applicable to well-surveyed areas. Conversely, the second run simulates a low confidence scenario applicable to legacy data or early stages of exploration. Disturbance distributions in the high input data confidence scenario were chosen to be of the Gaussian type with relatively low dispersion, whereas Uniform type distribution parameterized with large ranges were used for the low input data confidence scenario (Table 2).





**High input data confidence run**

For this run, a global information Entropy UIM voxel was produced to serve as a reference against matching topology-based estimates. Three vertical N-S cross-sections were extracted from the voxel at 250m, 500m and 750m Easting (Figure 10). The 250m and 750m information Entropy cross-sections are almost identical because the original model is symmetrical about the N-S median

cross-section where all structural data is located. Both sections display low to medium levels of Entropy (0.20 to 0.40) distributed around the original interfaces trace and forming Entropy halos of about 70m apparent thickness for non-triple-points areas. Conversely, triple points and areas of potential geometrical ambiguities display medium to high levels of Entropy (0.50 to 0.70) and thicker halos (~100m). The 500m information Entropy cross-section exhibits lower levels of Entropy and much thinner halos (~20m) because of its extreme proximity to the structural data inputs.

To verify topological stationarity, each plausible model was exported to a voxel that was used to build its corresponding adjacency matrix (Figure 6). Every "new" topology was placed into a standard topological stationarity graph (Figure 11). The number of distinct topologies observed over the process of generating plausible models appears to follow a logarithmic pattern. That is, the greater part of possible topologies are "discovered" quickly and further plausible model generation yields diminishing returns. In this instance, a third of topologies are discovered in a mere 3% of the total number of perturbations and the next third is completed

in under 25% of said number. The total number of observed distinct topologies represents about 5% of the total number of plausible models. Note that these finds are in accordance with previous work on topological stationarity in 3D geological modeling (Thiele et al., 2016b). Based on these observations, it is safe to assume topological stationarity for this run. Several parameter sets for DBSCAN were tested and it appeared that the only working set for this case is $\varepsilon = 0$ and $P_{min} = 1$. Otherwise, DBSCAN returns a single cluster along with a small number of unclustered topological signature. That is, each distinct topological signature has to

be considered as a cluster in itself in order to obtain more than one cluster. Such behavior is not entirely unexpected because of the low dispersion parameters set for the disturbance distributions. Indeed, low dispersion of disturbance distributions is partially and non-linearly correlated to low plausible model topological variability. This is confirmed by the low number (9) of non-null elements in the global internal information Entropy matrix (Table 3) which indicates that few topological relationships were affected by the perturbation process. With the aforementioned settings, DBSCAN returned fifty-five clusters that correspond to the fifty-five distinct

topological signatures present in the plausible models suite. A significance threshold of sixty occurrences was applied (Figure 12) to retain only the six most significant topological signatures and make subsequent steps more manageable, and such operation is only justified on the basis that topological stationarity is adequately met.

A representative plausible model was selected from each significant topological signature cluster and three vertical N-S cross-sections were taken (Figure 13) to obtain a qualitative view of the topological and geometrical differences between them. The 500m

Easting, median cross-section is mostly invariant throughout the cluster as pointed out by the low value observed on the global information Entropy UIM voxel (Figure 10). The 250m and 750m Easting cross-sections appear to be significantly more variable throughout the clusters in terms of distinct topological features and geometrical variations. Evident differences in section view include (i) the basin lower unit (Figure 13, green) gaining or losing contact with the metamorphic folded series (Figure 13, pinks) with the Mafic Cover separating the two series (Figure 13, blue), (ii) the basement (Figure 13, brown) coming into contact with the

mafic cover, (iii) the upper metamorphic folded unit (Figure 13, light pink) being in direct contact with the lower metamorphic unit (Figure 13, dark pink). Additionally, the potential traps highlighted in the original model are seen to change size and shape, to close and open throughout the clusters. These results indicate that topological signatures may help differentiate favorable scenarios in ore reservoir or oil and gas modeling applications.

Information Entropy cross-sections were extracted from the UIM voxels (Figure 10) that were generated for each significant

topological signature. Although, the information Entropy values look similar throughout the clusters, there are noticeable differences in terms of sharpness and triple-points differentiation. Predictably, the 500m Easting section shows very little extra-cluster variability and is very similar to its global counterpart. This is most likely because of its relative proximity to the original structural data inputs. In contrast, the 250m and 750m Easting sections display significant extra-cluster variability in terms of Entropy halos'



thicknesses (from 150m to 50m), triple-points differentiation (right ellipses) and sequence repetition in the metamorphic folded series (middle and left ellipses). As expected, cluster-based information Entropy cross-sections are all sharper than their non-clustered counterpart. This constitutes a strong indication that topological clusters are geometrically consistent and supports the thesis that topology is an efficient determinant for geological coherence. Additionally, sharper information Entropy cross-sections

imply sharper PGMs which allows for an increased external applicability of MCUE results. In general, these results underline the plausible model discriminating efficiency of topological signatures even when they are considered individually.

**Low input data confidence run**

As with the previous run, a global information Entropy UIM voxet was produced to serve as a reference against matching topology-based estimates. Equivalent cross-sections were taken (Figure 14) and exhibit very similar features to the high data confidence run.

However, attention is brought to the increased fuzziness of the information Entropy halos. These patterns can be explained by the disturbance distributions' selection and parameter selection for this run. The uniform distributions that were selected in this instance always have a higher innate Entropy compared to Gaussian distributions. Furthermore, the ranges selected largely exceed those of the previous run. Although at a lesser degree, the topological stationarity graph (Figure 15) expresses the same diminishing returns effect as the high input data confidence run. More specifically, a third of topologies were in the first 13% of plausible models,

another third in the next 20% of plausible models and the final third in the last 70% of plausible models. In this instance, DBSCAN was parameterized with $\varepsilon = 2$ and $P_{min} = 2$ and returned two topological signature clusters of size 953 and 39 respectively, along with 8 outliers. Lower or higher values for $\varepsilon$ and $P_{min}$ returned either a single cluster of size 1000 or a thousand clusters of size 1. Cross-sections extracted from representative models of both clusters (Figure 16) display stark differences at the geometrical and topological levels. Significant topological changes between the two clusters include the disappearance of the middle the

metamorphic folded unit (purple) from cluster 2, the emergence of the lower metamorphic folded unit (dark pink) against the lower basin unit (green) and the contact of the intrusion unit (red) with the upper metamorphic folded unit (light pink) in cluster 2. This is not surprising given the high number of non-null elements in the global internal information Entropy matrix (Table 4). Indeed, a total of twenty topological relationships were affected by the perturbation process to varying degrees. Moreover, per-cluster internal information Entropy matrices result in a significant number of non-null elements (Table ) which can be used to determine the main

"breaking" topological relationships when compared against each other and against the global matrix. Most topological shifts between the two clusters (red entries, Table 4) relate to internal topological relationships of the metamorphic folded unit and the basement. These shifts are consistent with the representative models' cross-sections and indicate that per-cluster internal information Entropy matrices may be used to draw geological inferences from their topological differences. When the clusters' internal Entropy matrices are compared against the global one, small differences become visible (underlined entries, Table 4) because of the inclusion

of the unclustered plausible models. Notably, the intermediate metamorphic folded unit entries are non-null against all other units and itself which suggests that the unit may be absent from some of the unclustered plausible models.

The information Entropy UIM cross-sections for cluster 1 shows little variation to its global counterpart (Figure 14). This is mainly due to the large size of cluster 1 compared to the number of plausible models. About 95% of plausible models carry a topological signature that links them to cluster 1. Given the convex nature of information Entropy, large clusters are likely to be near

undiscernible with the global population. Overall, cluster 2 displays sharper Entropy halos than cluster 1 or the global cross-sections. It also features strong aliasing because of its relatively small size (39). Information Entropy peaks about the metamorphic folded series appear to be shifted by a half of a fold wavelength between the two clusters (ellipses) while other features remain mostly constant. The relative similarity between both clusters information Entropy cross-sections (**Error! Reference source not found.**Figure 14) despite their strong geological, structural and topological disagreement suggests that topological clustering holds

potential as a differentiation tool in MCUE comparative analysis. Topological clustering would then be a way to mitigate the weaknesses of global information Entropy UIM in regard to structural relevance.





**Discussion**

In this paper, a basic procedure for topological clustering in MCUE was explored as possible improvement over currently available comparative analysis methods. The theoretical and practical aspects of the procedure were discussed and demonstrated over two proof of concept case studies.

The case for topological clustering rests on the fact that MCUE commonly generates geologically incompatible models because of the non-linear relationship between the plausible datasets and the plausible models' suite. This effect is introduced by the geometrical ruleset that implicit 3D geological modeling engines depend on to solve topological ambiguities. Ultimately, this topology-induced non-linearity translates into plausible model heterogeneity which is damaging to global comparative analysis methods that MCUE normally relies on and justifies topological clustering. Plausible model heterogeneity forms a strong logical

barrier to merging seemingly incompatible plausible models into a single PGM or UIM. Plausible models obtained through the perturbation of the same dataset may describe very different "realities" which correspond to different geological scenarios. Combining such incompatible model types that describe very different geological scenarios into a single uncertainty estimate is detrimental to the understanding of the quality of our knowledge in the area of interest.

Topological clustering provides more flexibility to external validation systems such as geophysical inversion or physical simulations

as it does not lock them into a single PGM or UIM. In turn, such approach holds the potential to make targeted groundtruthing easier as topological differences between clusters and per-cluster leverage analysis would help indicate which observations or topological relationships introduce heterogeneity in the plausible model suites. Furthermore, per cluster uncertainty is always lower than its global counterpart because of the convexity of UIMs. Therefore, topological clustering produces sharper per-cluster UIMs that are more comprehensible than the global UIM which helps to parameterize external validation systems. Topological clustering preserves

and improves geological knowledge since the differences between the topological signatures of distinct clusters are visible in the internal information Entropy matrices and can be interpreted in terms of geological relationships. Lastly, the proposed method increases the value of MCUE against analytical uncertainty propagation methods since the latter cannot consider the non-linearity that plausible model heterogeneity indicates. Analytical uncertainty propagation would estimate uncertainty from the interpolator directly without the need to build any more than a single PGM. However, it was shown that a single PGM cannot adequately express

the inherent non-linearity of the modeling engine. Note that this non-linear behavior is not a defect of the modeling engines themselves but rather a consequence of natural geological rules such as intrusion, cross-cutting or superposition.

Although promising, in its current form, the procedure may suffer from a number of limitations that concern DBSCAN and may indicate that other clustering algorithms such as k-means, c-means or machine learning are more appropriate. The low number of parameters, simplicity of the algorithm and low computational cost make DBSCAN an appealing choice for data clustering of large

datasets where the number and shape of clusters is unknown. However, DBSCAN suffers from a number of disadvantages that may hinder its ability to function effectively. The most relevant ones to this study are the "hidden" metric parameter, point density scale issues and conflicted points. The metric parameter relates to the choice of the metric used to compute de distance matrix. Datasets with high dimensionality may exhibit a degeneracy of the concept of distance when the data is uncorrelated and noisy. The issue is mostly covered by the fact that the topology of 3D geological model is usually well structured because of the geometrical ruleset's

influence. The point density scale issue relates to the spatial variance of point density throughout the dataset. A high spatial variance prevents an effective $\varepsilon$ parameterization because the concept of a reachable neighbor becomes ambiguous. In the case of basic topological signatures extracted from plausible models, the variability of the point density of clusters is usually low. That is so because the geometrical ruleset massively decreases the chances of odd topological signatures occurring. Note that this applies even for very low confidence disturbance distribution parameterization provided that all units are sufficiently informed. Conflicted points

relate to the fact that the DBSCAN algorithm is non-deterministic in some instances (Schubert et al., 2017). As a consequence, some border points may be reachable by several core points from different clusters at the same time. Although, DBSCAN only allows each point to belong to a single cluster. It is then the order in which the data was processed by the algorithm that will determine to which cluster these conflicted points belong to. For the purpose of this paper, this effect was avoided by parameterizing DBSCAN



with a low $\varepsilon$. Regardless of which clustering algorithm is chosen and how it is parameterized, the issue of the relevance of Boolean topological signatures clustering arises. Boolean topological signatures may be argued as being too simplistic in their representation of the actual geometrical relationships observed in the plausible model suites. Such oversimplification may inhibit the differentiating efficiency of the clustering algorithm. To address this problem, more accurate topological signatures may be used. The most

straightforward improvement is to distinguish normal and faulted contacts between geological units and express topological signatures as a ternary signal instead of a binary one. This solution is appealing because the rest of the procedure remains unchanged given that the Hamming distance is defined for all degrees.

Replacing lithological, unit-based, adjacency matrices with super, series-based, adjacency matrices is another possibility of improvement for the procedure. In this case, the geological units of a series would be considered as a single entry of the matrix. The

aim is to simplify the adjacency matrices, eliminate redundant information, decrease computational costs and increase readability. However, this approach assumes that series are topologically consistent which is not guaranteed as illustrated by the metamorphic folded series behavior in the low input data confidence run. In theory, better applicability of the procedure could be achieved by removing irrelevant topological relationships from the topological signature. The clustering algorithm would then be made blind to them and, in some cases, display higher differentiating ability. Although, the question of the relevance of a topological relationship

is likely to be ad hoc. At the practical level, in this paper, adjacency matrices were extracted from 3D grids obtained by discretizing the plausible 3D geological model. Therefore, adjacency matrices are prone to discretization artefacts when resolution is too low. Triple lines or triangulated interfaces could be used to derive the topological signatures while avoiding these artefacts.

Overall, more in-depth case studies are required to assess the capabilities of the method and determine the best route for possible improvements. More specifically, 3D real case studies are needed to better demonstrate the usability and practicability of the method

as opposed to the synthetic 2D section-based model used in this paper.

### Conclusion

In this paper, previous findings (Wellmann et al., 2014;Thiele et al., 2016a;Wellmann and Caumon, 2018) about plausible models' variability in MCUE were verified and a complete innovative comparative analysis procedure was proposed to address the issues raised by said findings. It was confirmed through experiment that MCUE outputs a significant proportion of geologically

incompatible plausible models and that topological analysis is a viable tool to differentiate them. The reasons for this incompatibility were discussed and were found to be due to the non-linear relationship between the plausible input datasets and the plausible models. That is, the model building process is non-linear itself. It was proposed that the model building non-linearity emanates from the geometrical ruleset that is used to constrain and partially define the topology of models in implicit 3D geological modeling engines. In view of this fact, topological clustering was proposed as a solution to distinguish geologically incompatible models. Therefore,

increasing the relevance and quality of the uncertainty indexes and probabilistic models in MCUE. Based off a two stages synthetic case study, it was found that topological analysis is a viable tool to differentiate geologically incompatible models and that topological signatures are strong indicators of geological features in 3D geological models. Topological analysis was shown to help reduce overall model uncertainty by ensuring topological consistency in the uncertainty indexes. Moreover, topology-driven comparative analysis may allow for higher model improvement potential than what standard uncertainty indexes or probabilistic

geological models allow for. The rationale is that improved knowledge of uncertainty allows users to target areas of interest where supplementary data collection is required to reduce said uncertainty. In this case, uncertainty is thought of as an improvement enabling tool that initiates a positive feedback loop and allows users to refine their understanding of the modelled area and increase the reliability of their model. This work finds applications in mining and oil & gas industries at the strategical and tactical stages of exploration or for mine development and planning. In particular, topologically consistent probabilistic geological models and their

associated topological signatures could be used as input for geophysical inversion and physical simulation software.



**Data/Code availability**

All datasets and models used in the present study are available online at https://doi.org/10.5281/zenodo.1202314.

**Appendices**

**Appendix A: The Spherical Cap distribution**

The spherical cap distribution is designed to describe variables that are uniformly distributed over any solid angle on the unit sphere $S^2$. The proposed parameterization is that of the mean/median direction spherical unit vector µ and half-aperture angle $\lambda$

(6)

$$p_{SC} = f(x|\mu, \lambda).$$

Start with the formula for the area of a spherical cap

10  (7)

$$A = 2\pi r^2 \big(1 - \cos(\theta)\big),$$

where $\theta$ is the polar angle and $r$ is the radius of the sphere. It ensues that, over $S^2$, the maximum value for $A$ is for $\theta = \pi$

(8)

$$A_{\max} = 2\pi \big(1 - cos(\pi)\big) = 4\pi.$$

The relative area of a spherical cap to the total sphere area is then given by

(9)

$$\frac{A_{\max}}{A} = \frac{2}{1 - \cos(\theta)}.$$

Given

(10)

$$\oiiint p_{SC} = 1,$$

And knowing

(11)

$$f(x|(.), \lambda = \pi) = \frac{1}{4\pi},$$

It follows that if

25  (12)

$$\mu^T x \geq \cos\lambda,$$

then

(13)

$$f(x|\mu, \lambda) = 4\pi^{-1} \frac{2}{1 - \cos(\lambda)}.$$

The authorized form is then

(14)

$$p_{SC}(x|\gamma, \lambda) = \begin{cases} \dfrac{1}{2\pi(1 - \cos\lambda)}, & \mu^T x \geq \cos\lambda \\ 0, & \text{else} \end{cases}.$$

**Appendix B: Spherical Cap pseudo random number generation**

To generate a Spherical cap uniformly distributed pseudo random spherical 3D unit vector $X_{\text{sphe}}$ on $S^2$ for a given mean direction

µ and range $\lambda$, define



$$(15)$$

$$X_{\mathrm{sphe}} = [\phi, \theta, r].$$

For $\mu = [0, (.), 1]$ the pseudo random vector is given by

$$(16)$$

$$X_{\mathrm{sphe}} = [\arcos(W), V, 1].$$

$W$ is given by

$$(17)$$

$$W = \cos(\lambda) + \xi,$$

where[†]

$$(18)$$

$$\xi \sim U(0, 1 - \cos(\lambda)).$$

$V$ is drawn as follows

$$(19)$$

$$V \sim U(0, 2\pi).$$

$X_{\mathrm{sphe}}$ should then be rotated to be consistent with the chosen $\mu$.

### Appendix C: Spherical standardized Irwin-Hall distribution

The standardized Irwin-Hall (IH) distribution is the distribution of the sum of a number of standardized uniformly distributed independent random variables

$$(20)$$

$$X = \sum_{i=1}^{n} U_n,$$

with all $U_n$ drawn from $U(-a, a)$. This distribution is useful in Bayesian inference as it models the sequenced hypersampling of a standardized uniform distribution in a compact form. For $a = \frac{1}{2}$, the IH distribution density is given by

$$(21)$$

$$f_X(x|n) = \frac{1}{2(n-1)!} \sum_{i=0}^{n} (-1)^i \binom{n}{i} \left(x + \frac{n}{2} - i\right)^{n-1} \mathrm{sign}\left(x + \frac{n}{2} - k\right).$$

In this form, its mean is always 0 and variance is $\frac{n}{12}$. The standardized IH distribution can be redefined as the chain convolution of its uniform components. For example,

$$(22)$$

$$f_X(x|n = 2) \equiv U(-a, a) * U(-a, a).$$

Using the convolution theorem, this can be generalized to

$$(23)$$

$$f_X(x|n) \propto \mathcal{F}^{-1}\left(\mathcal{F}(U(-a, a))^n\right),$$

where $\mathcal{F}$ is the Fourier transform and $\mathcal{F}^{-1}$ its inverse. Substituting (9) into (18), one finds that the standardized spherical IH distribution of order $n$ is proportional to the inverse Fourier transform of the $n$-exponentiated Fourier transform of the standardized Spherical cap distribution

$$(24)$$

$$S_{\mathrm{IH}}^n \propto \mathcal{F}^{-1}\left(\mathcal{F}(p_{\mathrm{SC}}(x|[0, (.), 1], \lambda))^n\right),$$

with

---

[†] $U(a, b)$ is the usual continuous uniform distribution.





$$(25)$$

$$\mathcal{F}\big(p_{SC}(x|\gamma, \lambda)\big) = \frac{sin\left(\frac{\pi\omega}{2}\right)}{\sqrt{2\pi}\big(\pi\omega - \pi\omega cos(\lambda)\big)}.$$

**Competing interest**

The authors declare that they have no conflict of interest.

5      **Funding**

Funding: This work was supported by the Geological Survey of Western Australia; the Western Australian Fellowship Program;
and the Australian Research Council for their financial support

**Acknowledgements**

The authors would like to thank Intrepid Geophysics for their participation in the software development effort that proved essential
10    to the completion of this project.





**Figures**

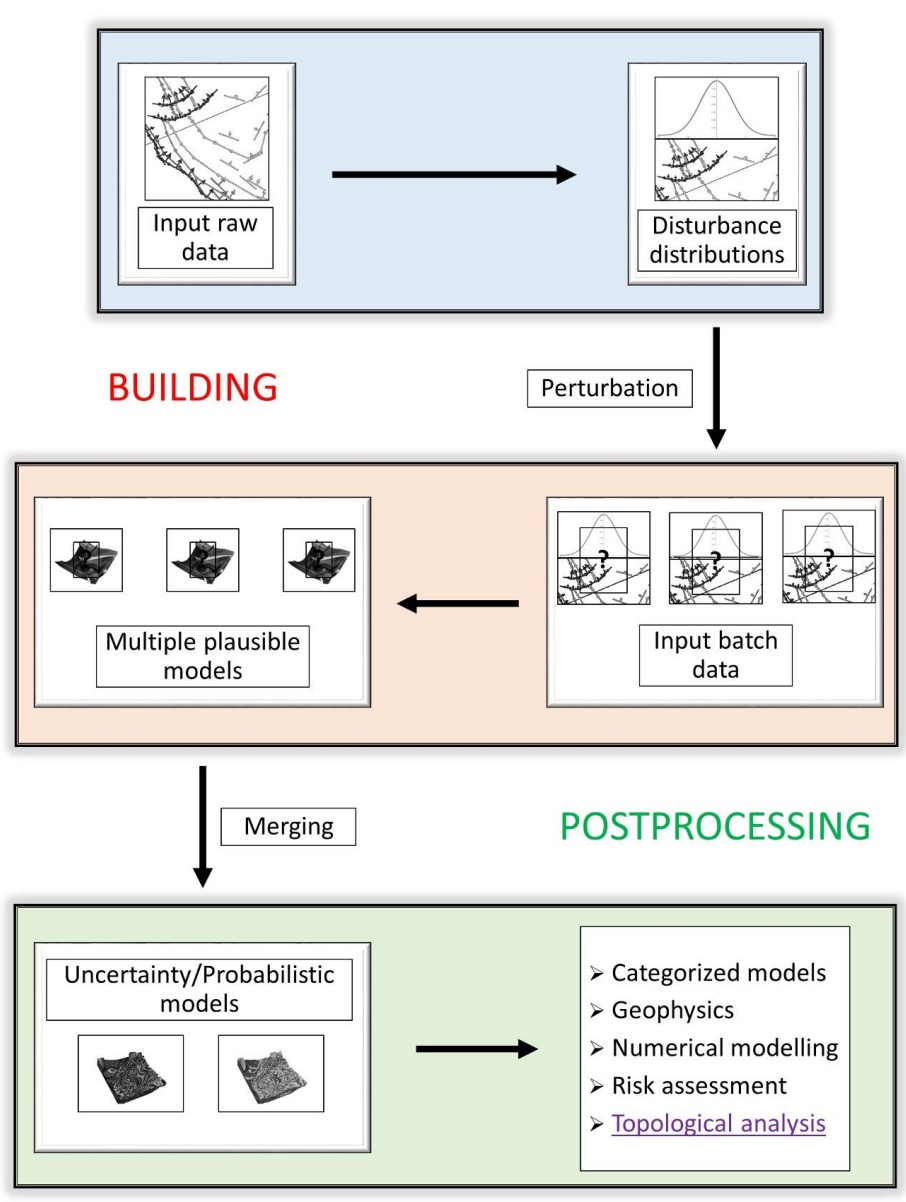

**Figure 1 MCUE simplified procedure, Modified from Pakyuz-Charrier et al 2018**



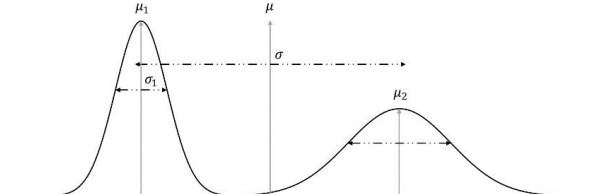

**Figure 2 Bimodal distribution with associated global and modal dispersion parameters**



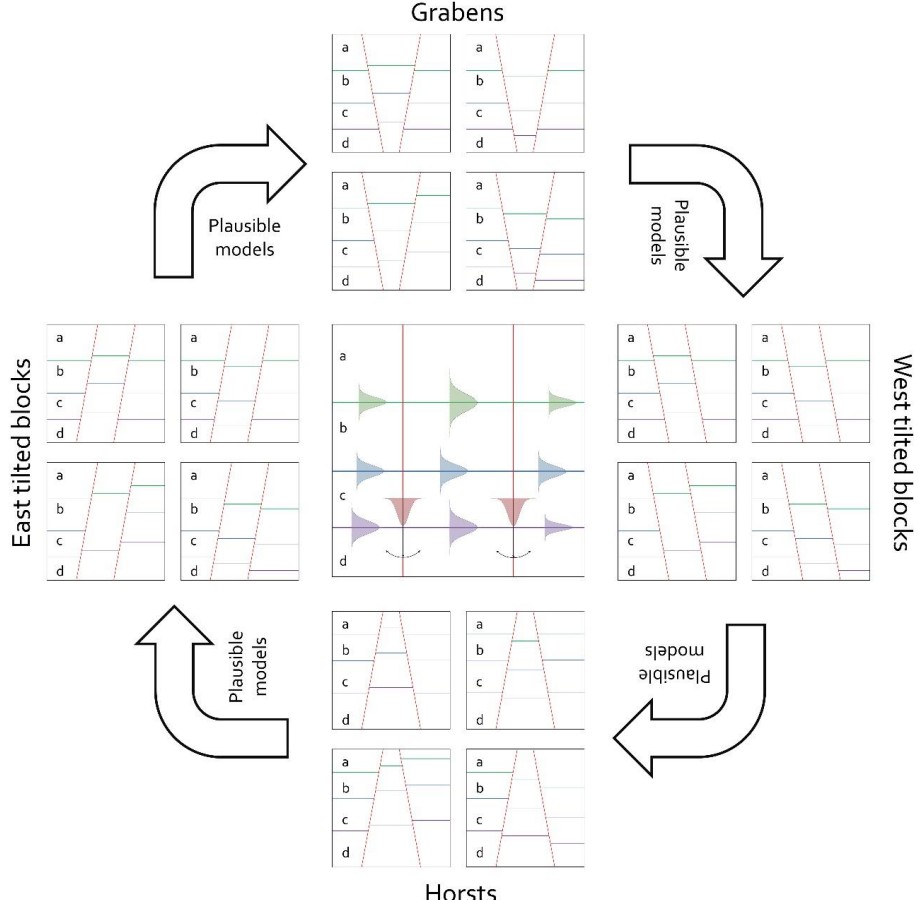

**Figure 3 Example of equally plausible yet geologically incompatible models**



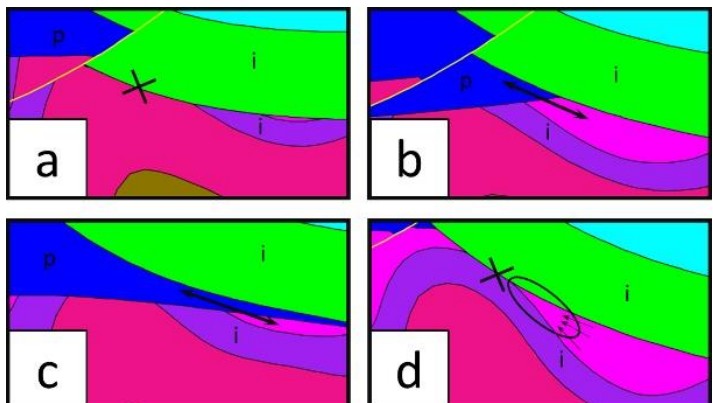

**Figure 4 The open or closed status of an ore deposit sedimentary trap varies with the topology of surrounding impermeable (i) units.**



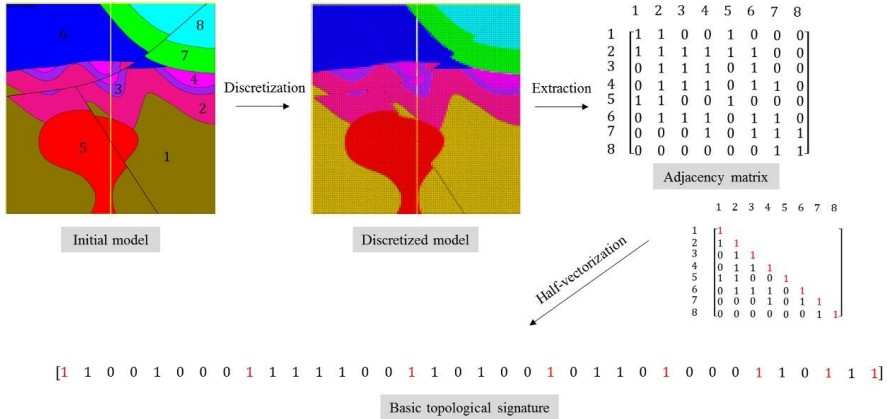

**Figure 5 Procedure for topological signature extraction**





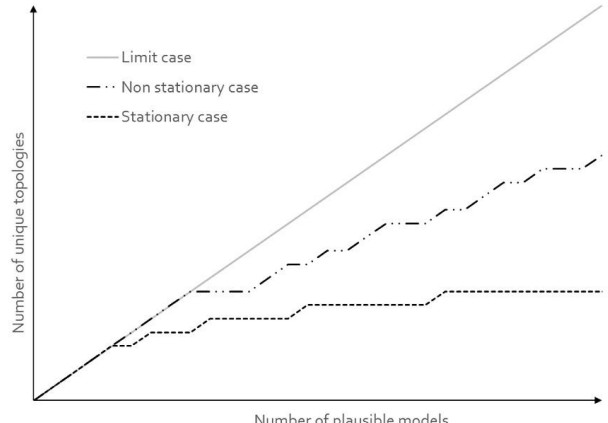

**Figure 6 Topological stationarity graph with example cases**





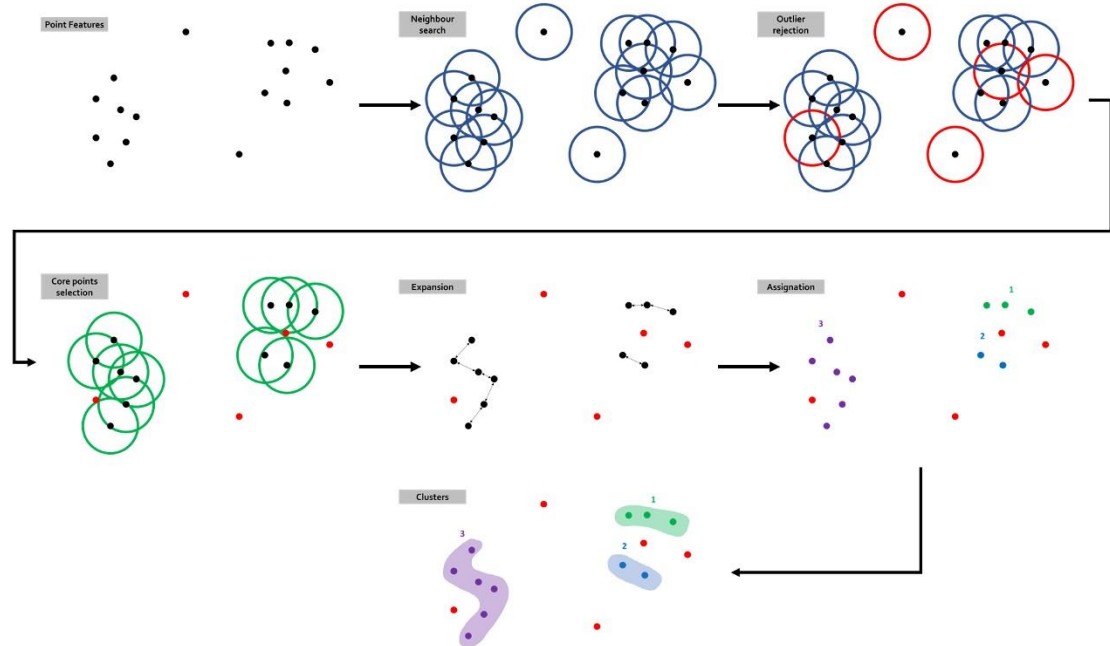

**Figure 7 Density-Based Spatial Clustering of Applications (DBCAN) workflow**





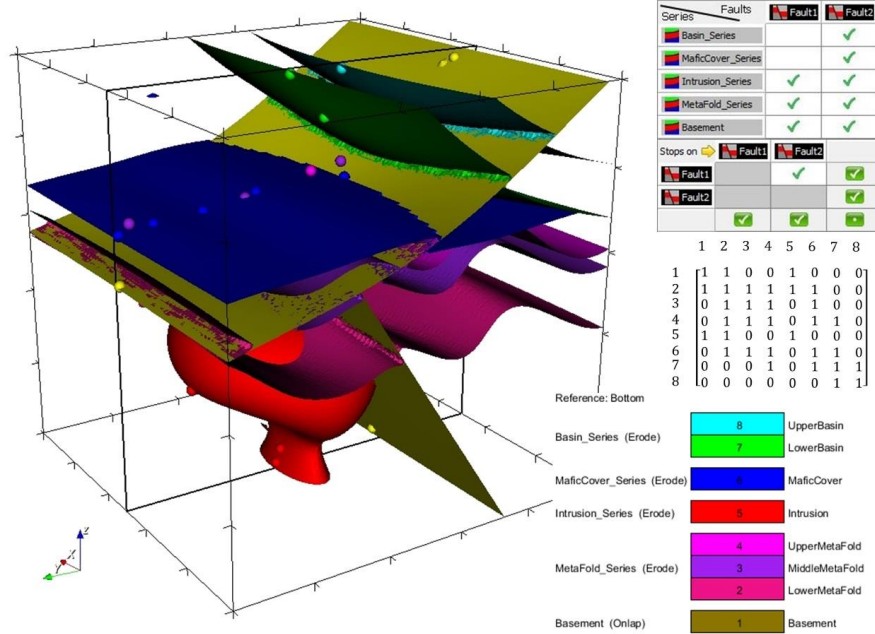

**Figure 8 CarloTopo 3D geological model with original input foliations (disks) and interfaces (points), geometrical rulesets for units and faults, and adjacency matrix. The model box is kilometric and all data is on the x=500m vertical cross-section**




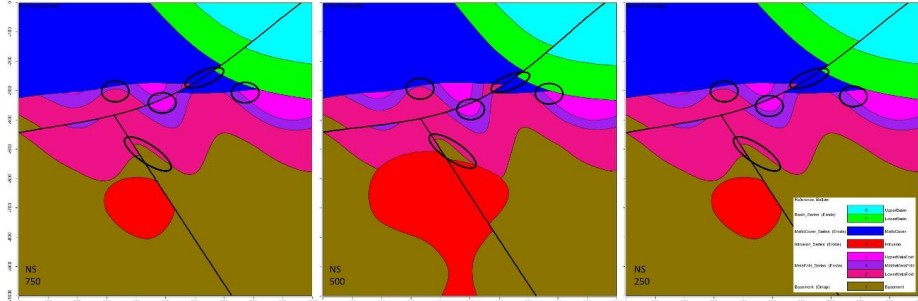

**Figure 9 Original CarloTopo vertical cross-sections at x=250m, 500m and 750m with potential ore deposit traps or channels circled**



**Figure 10 Global (top row) and top 5 most significant topological signatures vertical cross-sections of information Entropy uncertainty index models (UIM) for the low input data confidence run**



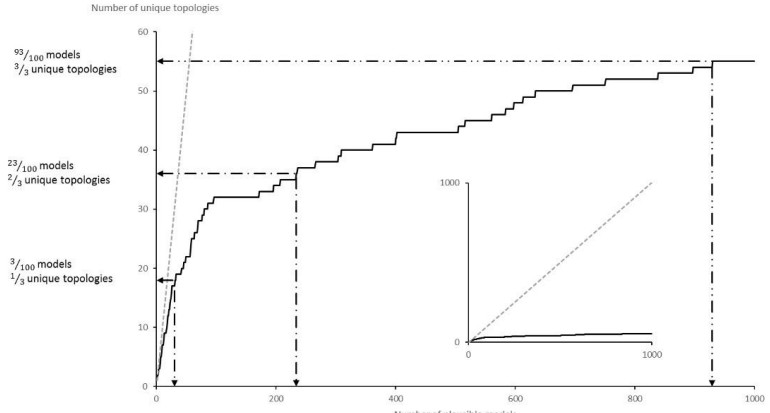

**Figure 11 Topological stationarity graph for the CarloTopo high input data confidence run. 1:1 graph in background as reference**




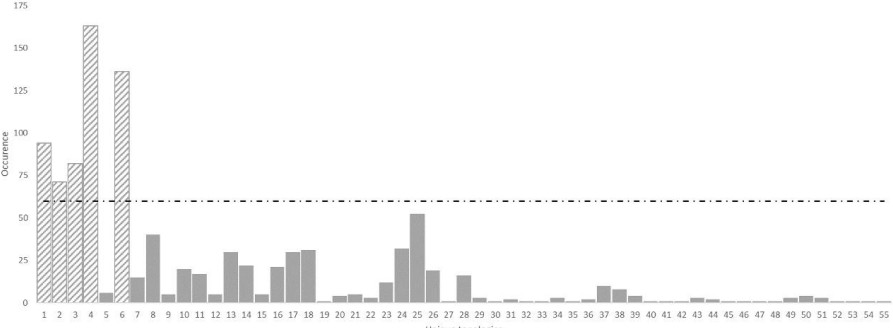

**Figure 12 Unique topologies occurrences for the high input data confidence run with significance threshold of 60. Note that in this instance, the clustering algorithm returned every topological signature as a distinct cluster**



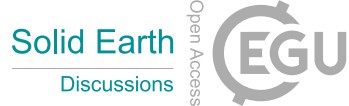

**Figure 13 Vertical cross-sections of representative plausible models for the top 5 most significant topological signatures in the high input data confidence run. Major topological changes are circled**





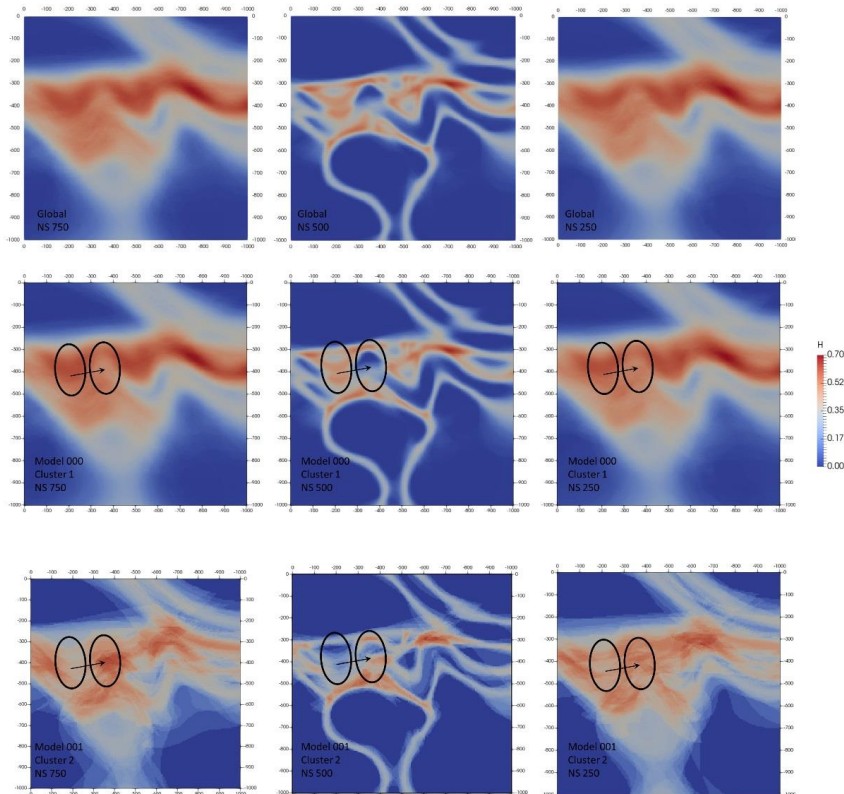

**Figure 14 Global (top row) and per-cluster vertical cross-sections of information Entropy uncertainty index models (UIM) for the low input data confidence run**





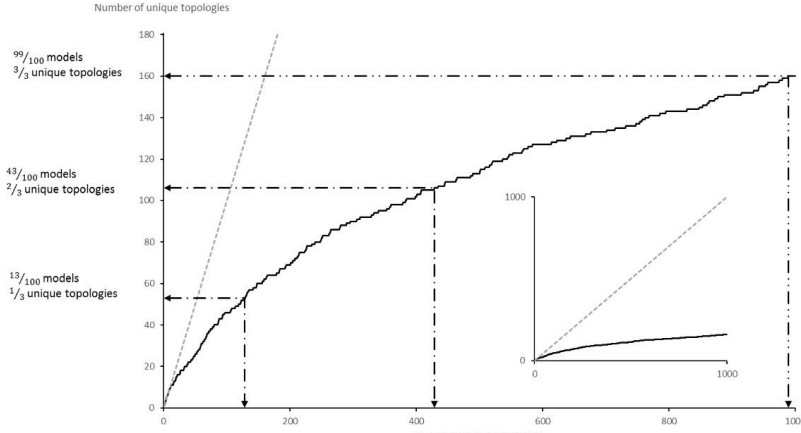

**Figure 15 Topological stationarity graph for the CarloTopo low input data confidence run. 1:1 graph in background as reference**





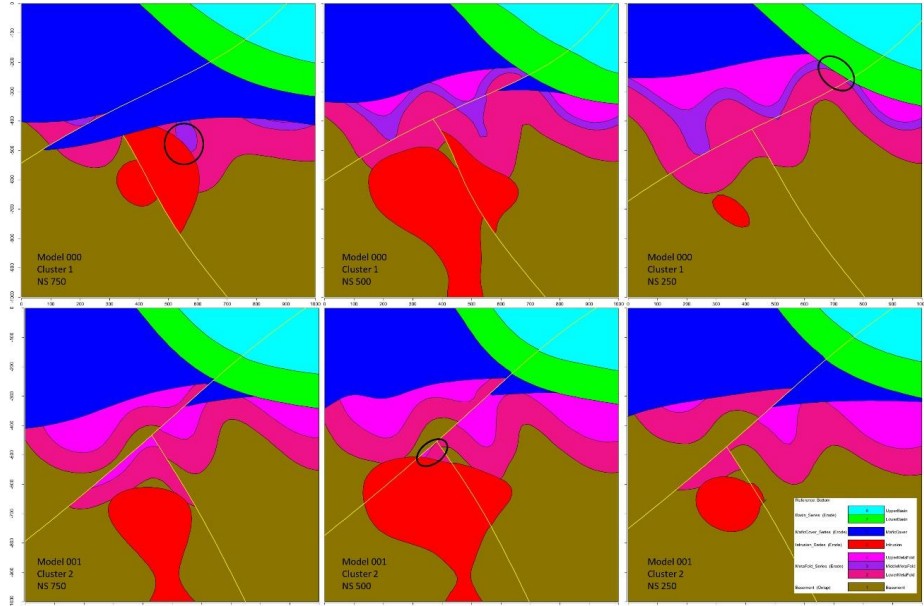

**Figure 16 Vertical cross-sections of representative plausible models for each cluster in the low input data confidence run. Major topological changes are circled**



**Tables**

**Table 1 Original input structural data description for the CarloTopo 3D geological model**

|  | #Foliations | #Interface points |
|---|---|---|
| UpperBasin | 1 | 4 |
| LowerBasin | 1 | 4 |
| MaficCover | 1 | 7 |
| Intrusion | 8 | 11 |
| UpperMetaFold | 1 | 2 |
| MiddleMetaFold | 6 | 7 |
| LowerMetaFold | 5 | 6 |
| Basement | NA | NA |
| Fault1 | 1 | 3 |
| Fault2 | 1 | 2 |
| Total | 25 | 46 |



**Table 2 Summary of all MCUE parameters used in this study**

| CarloTopo Model | | |
|---|---|---|
| | High confidence scenario | Low confidence scenario |
| Foliation orientation perturbation parameters | $vMF(\zeta,50)^{\#}$ | $SC(\zeta,20°)$ |
| Foliation/Interface location perturbation parameters | $N(\varepsilon,5m^{*})$ | $U(\varepsilon,25m^{**})$ |

$^{*}$ 0.5% of box extent  
$^{**}$ 2.5% of box extent  
$^{\#}$ 16° 95% confidence interval

SC: Spherical cap distribution  
vMF: von Mises-Fisher distribution  
U: Continuous uniform distribution  
N: Normal distribution  
$\zeta$: original foliation  
$\varepsilon$: original interface point





**Table 3 Global internal information Entropy matrix for the high input data confidence run**

| | High Confidence Run Global Entropy | | | | | | | |
| | 0 | 1 | 2 | 3 | 4 | 5 | 6 | 7 |
|---|---|---|---|---|---|---|---|---|
| 0 | 0.00 | | | | | | | |
| 1 | 0.00 | 0.00 | | | | | | |
| 2 | 0.50 | 0.00 | 0.00 | | | | | |
| 3 | 0.34 | 0.17 | 0.00 | 0.00 | | | | |
| 4 | 0.00 | 0.00 | 0.51 | 0.41 | 0.00 | | | |
| 5 | 0.37 | 0.00 | 0.00 | 0.00 | 0.00 | 0.00 | | |
| 6 | 0.08 | 0.52 | 0.28 | 0.00 | 0.00 | 0.00 | 0.00 | |
| 7 | 0.00 | 0.00 | 0.00 | 0.00 | 0.00 | 0.00 | 0.00 | 0.00 |



**Table 4 Per-cluster (top), global (bottom left), and contrast (bottom right) internal information Entropy matrices for the low input data confidence run**

### Low Confidence Run Cluster1 Entropy

|   | 1 | 2 | 3 | 4 | 5 | 6 | 7 | 8 |
|---|---|---|---|---|---|---|---|---|
| 1 | 0.00 | | | | | | | |
| 2 | 0.00 | 0.00 | | | | | | |
| 3 | 0.53 | 0.00 | 0.00 | | | | | |
| 4 | 0.45 | 0.18 | 0.00 | 0.00 | | | | |
| 5 | 0.00 | 0.00 | 0.49 | 0.52 | 0.00 | | | |
| 6 | 0.22 | 0.00 | 0.00 | 0.00 | 0.25 | 0.00 | | |
| 7 | 0.30 | 0.31 | 0.13 | 0.00 | 0.10 | 0.00 | 0.00 | |
| 8 | 0.00 | 0.00 | 0.01 | 0.03 | 0.00 | 0.20 | 0.00 | 0.00 |

### Low Confidence Run Cluster2 Entropy

|   | 1 | 2 | 3 | 4 | 5 | 6 | 7 | 8 |
|---|---|---|---|---|---|---|---|---|
| 1 | 0.00 | | | | | | | |
| 2 | 0.00 | 0.00 | | | | | | |
| 3 | 0.00 | 0.00 | 0.00 | | | | | |
| 4 | 0.20 | 0.00 | 0.00 | 0.00 | | | | |
| 5 | 0.00 | 0.00 | 0.00 | 0.29 | 0.00 | | | |
| 6 | 0.00 | 0.00 | 0.00 | 0.00 | 0.22 | 0.00 | | |
| 7 | 0.45 | 0.34 | 0.00 | 0.00 | 0.00 | 0.00 | 0.00 | |
| 8 | 0.00 | 0.00 | 0.00 | 0.00 | 0.00 | 0.14 | 0.00 | 0.00 |

### Low Confidence Run Global Entropy

|   | 1 | 2 | 3 | 4 | 5 | 6 | 7 | 8 |
|---|---|---|---|---|---|---|---|---|
| 1 | 0.00 | | | | | | | |
| 2 | 0.00 | 0.00 | | | | | | |
| 3 | 0.53 | 0.06 | 0.06 | | | | | |
| 4 | 0.48 | 0.18 | 0.06 | 0.00 | | | | |
| 5 | 0.00 | 0.00 | 0.50 | 0.52 | 0.00 | | | |
| 6 | 0.21 | 0.00 | 0.06 | 0.00 | 0.25 | 0.00 | | |
| 7 | 0.31 | 0.31 | 0.17 | 0.00 | 0.10 | 0.00 | 0.00 | |
| 8 | 0.00 | 0.03 | 0.04 | 0.06 | 0.01 | 0.22 | 0.00 | 0.00 |

### Low Confidence Run Cluster Entropy Absolute Difference

|   | 1 | 2 | 3 | 4 | 5 | 6 | 7 | 8 |
|---|---|---|---|---|---|---|---|---|
| 1 | 0.00 | | | | | | | |
| 2 | 0.00 | 0.00 | | | | | | |
| 3 | 0.53 | 0.00 | 0.00 | | | | | |
| 4 | 0.25 | 0.18 | 0.00 | 0.00 | | | | |
| 5 | 0.00 | 0.00 | 0.49 | 0.23 | 0.00 | | | |
| 6 | 0.22 | 0.00 | 0.00 | 0.00 | 0.03 | 0.00 | | |
| 7 | 0.14 | 0.03 | 0.13 | 0.00 | 0.10 | 0.00 | 0.00 | |
| 8 | 0.00 | 0.00 | 0.01 | 0.03 | 0.00 | 0.07 | 0.00 | 0.00 |




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
