# Peer review of "Topological Analysis in Monte Carlo Simulation for Uncertainty Propagation"

_Solid Earth, 2019_

## Referee Comment (RC1) · Guillaume Caumon (Referee) · 11 Jun 2019

**1   General comments**

This paper proposes and evaluates a method to define clusters in a population of structural models obtained by spatial data perturbation. An original feature of the approach is that the clustering uses only topological distances. I would like to congratulate the authors for addressing this difficult problem, which is very relevant uncertainty management in structural studies.

The method builds mainly on previous papers by Thiele et al (2016) and on the DB-SCAN algorithm. The evaluation is made on two populations of models obtained with

varying degrees of data uncertainty. The results are very interesting, as the proposed method does identify relevant population subsets, but only very few clusters manage to be identified even when the method's parameters are tuned. This suggests the structure of the problem is very continuous, and tends to generate models which are mutually very close one to another when considering the adjacencies of geological units.

I mainly have form comments, which I hope will will help to improve the paper.

**2   Acronyms and wording**

Overall, the paper is well written and easy to read, but heavily uses three acronyms: MCUE, UIM, PGM, which I have not seen in other author's work working on similar topics. So, I wonder if we really need these acronyms.

In particular, the term MCUE (Monte Carlo Simulation for Uncertainty Estimation) is very general and not specific to the proposed method, so I'd recommend to change the name to better explain that the data are perturbed / sampled to generate a set of probabilistic geological structural models.

PGM (Probabilitic geological models) is quite clear, but I am not 100

UIM (Uncertainty Index Models) is expanded in the introduction, but not explained before page 3 (mention to the work of Wellmann and Regenauer-Lieb, 2012), so could be difficult to understand upfront. Why not just mention a map of local uncertainty?

Another point of vocabulary is the term "geologically incompatible models" and "geologically [in]consistent". To me, geologically incompatible would mean transforming a reverse fault into a normal fault, or changing the geological history. I understand that the proposed data perturbation may change a normal fault, but this is not what the clustering detects. In this paper, consistency / compatibility essentially means topolog-

ical similarity, so I'd rather use that term. Of course, topological differences may have implication in the geological history (e.g., the juxtaposition of one formation against another may create paths for subsurface to migrate), but this is not mandatory. Geometrical variability between models of the same topology could have a similar effect, see for example Abrahamsen et al (2015). I would recommend to stay with descriptive terms (topologically similar / dissimilar) in the bulk of the manuscript and discuss the geological implications in the case studies and in the discussion.

Overall, I think the paper could be improved by more thoughly citing and refering works on structural uncertainty done by other teams. In Section 4 of Wellmann and Caumon (2018), we tried to review the various approaches to structural uncertainty assessment, so I hope this could be a useful entry point.

**3  Motivation**

The introduction in its present form presents the problem in a very general way, which is nice... but maybe too general. I think the intro could do a better job to motivate the need for model clustering, which is the key aspect of this paper. This goes along the lines of clarifying (or replacing) the term "incompatibility", and possibly also of explaining a bit more in what sense a categorization of models could help to reduce uncertainties (as mentioned at the beginning of section 3). This would certainly call for some additional references to inverse problem theory, which can look around a particular scenario in model parameter space, but has more difficulties to work with problems of varying numbers of parameters. Carter et al (2006); Suzuki et al (2008), Cherpeau et al (2012), Scheidt et al (2018) could be useful references to discuss this.

**4   Heteroscedasticity and error correlation**

In Section 1.2 and other places, heteroscedasticity in the data set is invoked to imply dependency within the data. I agree, but this is not the only reason. Multiple examples of spatial correlation have been documented in the literature, especially when the data are interpreted from seismic images (where location errors stem from velocity errors). Although rare, this could also occur in principle with field geological data (e.g., poorly calibrated instrument leading to systematic measurement bias in some areas). At the bottom of page 4, the authors seem to suggest that heteroscedasticity always implies spatial correlation. I am not sure whether this is correct and would argue that heteroscedasticity and spatial correlation are two different (and important) aspects of data uncertainty.

**5   What does topology exactly means?**

If I understood correctly, the type of topology used in this paper is primarily "Lithological topology" (sensu Thiele et al., 2016), ie, the nodes of the topological graph represent faulted, folded and possibly eroded geological formations while the edges of the graph represent the adjacency between these formations. This is clear from the title of section 3.1, but it was not clear to me when I first read the paper, which trigerred many interrogations. Having now understood that lithological topology is considered, I still have two comments:

- Considering 1's on the diagonal seems like a choice that a formation is considered adjacent to itself (although one could argue that this is not really adjacency). Did I miss something here? Actually, I do not think the daigonal is so important in the characterization, so would it make sense to just ignore the diagonal in further steps?

- The "lithological topology" considers only geological formations and not the connected components of these formations (termed "cellular topology" by Thiele et al.), whose number may change from one stochastic structural model to the next. So, the existence of the same number of lithologies in all structural models seem like a prerequisite to apply the proposed method. This should be more explicit. The variability in lithologies only summarizes much of the variability that would be observed considering the adjacencies of connected components. I suspect that this contributes to the reason why the clustering algorithm has difficulties to segragate realizations.

**6  Cluster Entropy**

I am not completely sure I understand the cluster entropy concept, because I suspect there is a typo in the equation: I am not sure about the k in the log, and it seems to me that (if k is indeed a mistake), the result will always be zero (sum of 0 log(0) and 1 log (1)). I suspect there should be some average connectivity involved (probably $\frac{A(k)_i^j}{c} log \left( \frac{A(k)_i^j}{c} \right)$ and not just $A(k)_i^j log \left( A(k)_i^j \right)$). Maybe I am just missing something, but in any case some references would be welcome.

**7  Minor remarks**

- page 3: "flattened to triangulated surfaces or shrink to triple lines": Unclear to me.

- page 5: the mention to adjacency, overlap and separation are already made in Thiele et al. (2016), and only adjacency is used in this paper, so maybe there is

no need for discussing the combinatorial aspects here.

- It took me som guesswork to unsrstand Table 4. Please explain that the 1-8 codes correspond to lithologies; having the table of lithologies would help analyzing the results and following the discussion. The lower right matrix is the difference of the matrices in the first raw, right? Please add "See text for detail" in the caption, not all elements are described in the caption.

- Considering the most significant topological classes (page 8) is acceptable but it is arguable for uncertainty quantification is high dimensional spaces, as it may artificially reduce uncertainties. I think this should be mentioned.

- Some of the discussion on distances could possibly benefit from references to the recent book of Scheidt et al (2018).

- The Appendix provides interesting details about spherical orientation

Please see also annotated manuscript.

[Figure]

**Supplement:**

[revised manuscript text omitted]

---

## Referee Comment (RC2) · Gautier Laurent (Referee) · 9 Jul 2019

The manuscript entitled "Topological Analysis in Monte Carlo Simulation for Uncertainty Estimation" proposes to combine mathematical description of topology as presented by Thiele et al. (2016) to Monte Carlo approaches to geological uncertainty estimation as presented by Wellman (2013) and Lindsay et al. (2013). In particular, the authors demonstrate the use of DBSCAN clustering algorithm for highlighting the main topological families.

The proposed approach provides very interesting complements for this overall uncertainty estimation approach. It's interest lies in the fact that topological changes in produced models often have a first order impact on resulting physical simulations (eg., for

fluid flow simulations in reservoir modelling). It is only natural to use topology as a metric for clustering the sampling outputs.

However, there are several limitations to the proposed approach, which should not prevent its publication but rather call for further discussion:

- First of all, this manuscript inherits from the limitations of the approach to uncertainty sampling that consist in varying the input data independently. This is making the assumption that uncertainty is mainly due to measurement errors, while the matter of data representativeness and lack of knowledge account for a probably larger part in the geological uncertainty. (cf., discussion below)

- A second limitation comes from the use of adjacency matrices as a unique measure of topology. This is quite efficient indeed, as demonstrated in this manuscript, but this metrics is unable to completely account for topology. As a matter of fact, two models that are not homeomorphic could have the same signature (cf. attached drawings), which limits the justification for calling it a signature. This should be better discussed. As a consequence, considering your remark at p8L37 "These results indicate that topological signatures may help differentiate favourable scenarios in ore reservoir or oil and gas modelling applications." most of these traps are actually invisible in the topological signature whereas they represent peculiar topological features. This is a serious limitation in my opinion and should be discussed.

Besides these drawbacks, the manuscript is relatively well written and illustrated, even if the figure captions should be more informative. Some recurrent grammatical errors are still making the reading difficult in places. Avoiding the inadequate use of possessive apostrophes should greatly improve the reading. Acronyms that are not commonly used in the community are used throughout the document, which makes it difficult to follow the discussion in place.

Based on these appreciations and the detailed comments below, I would suggest that major revisions should be made before accepting this manuscript for publication.

**1 Detailed comments**

1.1 Lack of explanation of some basic concepts

- p2L.9: produce probabilistic geological models (PGM) and uncertainty index models (UIM). should be defined

- p2L. 15 "Plausible models' incompatibility is damaging to the relevance of MCUE because the PGMs and UIMs implicitly assume plausible model homogeneity." -> I don't understand this statement. What is this assumption for homogeneity?

1.2 The source of sampled uncertainty

- p2L.39: these are only a (probably small) part of the uncertainty. Problem of representativity is much bigger for data points, and what about other type of uncertainties

- p3L7: the problem is that simplifying the source of uncertainty as a measurement error make the following statement possible "Sampling is usually made independently as errors do not show any spatial dependency, because measurements are physically independent (Pakyuz- Charrier et al., 2017b).", which is hiding the fact that different measurements may be sampling a same structure thus making the measurements physically dependent. This situation seems closer to a rule than an exception for geological structures, on the contrary to what is suggested by the following paragraph (limiting it to cyclicity or observed heteroscedasticity).

Although disturbance distribution is not the main topic of this paper, this limitation to the method should be further discussed and stated more clearly.

- p4L31: "Moreover, these methods work best if errors are spatially dependent. This is normally not the case for sparse geological structural measurements taken individually. Actually, there is no logical reason to consider that the measurement errors related to, for example, two foliations measured with a compass in different areas are dependent on one another." Is it equivalent to say measurement errors are independent and the uncertainty about the measures are independent? If two measures are informing about a same structure, then one is also providing information about the other measurement and they are not independent. Consider an uncertain measurement A dipping 15 degree to the north with a possible error of + or - 5 degrees, if you add another measure B dipping 5 degree to the north + or - 5, you are probably more likely to have a somewhat flat layer dipping 10 degrees to the north than to have a large fold structure. Therefore the errors should not be looked at independently. It does not mean you can not make this assumption to simplify the mathematical treatment of a dataset, but you can not justify it by stating that uncertainties are independent.

**1.3 Unimodality**

- p4L9: I don't see why computing stratigraphic variability or information entropy would have any assumption about unimodality?

- p4L10-14: entropy and stratigraphic variability are scalars that inform about the variability in a set of models. They are not meant to reconstitute a whole distribution. It is not sufficient information for a bimodal distribution but it would not be for a general unimodal distribution as well. I don't see your point and your argument should be clarified.

- p4L16: how do you define a "homogenous population of plausible models" or heterogeneous (P4L19)

**1.4 The discussion about linearity seems confused**

- p4L21 : the concept of non-linearity that you are using here is not clearly explained. On one side you are using linearity in terms of the mathematical combination of data values (p4L22) but the non-linearity that is then discussed is based on the geometrical and more particularly the topological continuity of the discussed changes. Of course, modifying the geological ruleset introduces more extreme changes in the model and would certainly impact its topology, but simple (linear) perturbation of a given dataset may also result in topological changes even with the same geological rules applied. This is one of the disadvantages of implicit approaches vs. explicit ones. You could refer to Collon Caumon 2017 for these aspects: DOI: 10.3997/2214-4609.201701144 I would suggest to clarify what you describe as a linear change and to rely on topological terms such as the concept of homeomorphism here.

- p4L27: (ii) plausible dataset variography is not a reliable indicator of plausible model homogeneity. Why would that be? What is your point?

**1.5 Topological signature**

- Fig. 5: why discretizing, boundary model (eg. surface model) should contain enough information?

- how are faults described? It looks like they are not whereas this is a major aspect in model topology.

- the diagonal of 1 is useless and should be removed for compression. Actually, later in the manuscript, with the second example we realise that this is actually encoding the presence of the layer. This should be explained more clearly.

- P6L9-10 : This is not clear: "Note that clustering the topological signatures of the plausible model suite implies that quantitative topological stationarity is not required." To me it seems that clustering doesn't required stationarity, but this is not what this sentence is saying.

**1.6 Entropy**

- eq. 3: why is there a k factor inside the log parenthesis?

**1.7 Post-process**

- p7L7: Central statistics: this term should be explain instead of calling it straight-forward This process is very interesting but its implementation needs to be better explained In addition, this process is not demonstrated in the examples. Please either detail or remove.

- p8L25: the threshold of 60 is not justified. Could you discuss this choice?

**1.8 References**

- inconsistency of reference ordering, should generally be alphabetically or chrono-logically, e.g. P2L1-2: "Schweizer et al., 2017;Wang et al.,2016;Nearing et al., 2016;Aguilar et al., 2018;Mery et al., 2017;Dang et al., 2017;Lark et al., 2013"

- p2L13. and p4L18 (Thiele et al., 2016a;Thiele et al., 2016b) should be gathered into (Thiele et al., 2016a,b)

- P4L15: Pakyuz-Charrier et al., 2018;Pakyuz Charrier et al., 2018 : he same reference is repeated twice

- It seems that the three appendices are not referenced in the main text.

1.9   Grammar remarks and other miscellaneous remarks

- p3L22-23: the construction of the sentence and the enumeration are making it difficult to read.

- p3L35: the concept of using triple lines for comparing models require a reference or further explanation

- p4L2: groundtruthing doesn't look like a formal english word and should be explained anyway.

- p4L9-11: These two sentences are repeating the same information, I suggest to remove the first one which is less detailed. "The underlying assumption is that the plausible models constitute a unimodal population and may be analyzed as such. The UIM used in MCUE are scalar proxies for categorical uncertainty and one of the critical conditions for a single scalar to be representative of the uncertainty of a variable is that it has to be distributed unimodally."

- p4L29: remove "are"

- in the whole manuscript: authors are making an extensive use of possessive apostrophes in forms than are grammatically incorrect after eg. https://www.ef.com/wwen/english-resources/english-grammar/forming-possessive/

- p6L2: there is a double dot at the end of the line

- p6L4: there is no 's' at the end of signature in "Cumulative observed topological signatures graphs"

- p6L17: each points -> each point

- P6L24: implementation wise -> implementation-wise

- p6L39: (3 -> Eq. (3)

- p7L4: incorrect grammar construction

- p7L11: central, -> central statistics,

- p7L33: "with a," remove the coma

- p8L39: Error! Reference source not found.

**1.10   Image quality**

- The quality of fig. 3 is poor (to the point it is difficult to identify colors )

- The quality of fig 4 should be improved as well.

- The figures are not sufficiently described and rely to much on the text for explanations. For example, the caption should state why annotation such as ellipses are made for.

**1.11   Fig 3**

- is lacking description and explanation

- there are misleading geological terms: for the models labeled grabens, only the one in the top right corner is actually a graben, similarly for the horsts, only the top left one is actually a horst the different tilted blocs are in fact not tilted and a layer is missing for the two model in the lower left corner of the tilted models.

- why are the arrows labelled plausible models?
* * *
[Figure]

[Figure]

[Figure]

| | A | B | C | D |
|---|---|---|---|---|
| A | 1 | | | |
| B | 1 | 1 | | |
| C | 1 | 1 | 1 | |
| D | 1 | 1 | | 1 | 1 |

[Figure]

[Figure]

| | A | B | C |
|---|---|---|---|
| A | 1 | | |
| B | 1 | 1 | |
| C | 1 | 1 | 1 |

**Fig. 1.**

---

## Author Comment (AC1) · 13 Aug 2019

Authors response to G. Caumon comments

1 General comments

The relative lack of variability is actually an effect of the initial parameterization of the co-kriging interpolator for the cover unit (blue) and folded units (pink). A strong variographic anisotropy was enforced on these units over the X axis. This was done to prevent excessive variations within the plausible model suite and ease interpretation of results. The paper was updated to reflect this point.

2 Acronyms and wording

[Figure]

>The UIM and PGM acronyms have been removed and replaced with their fully descriptive names.

>The MCUE acronym was replaced with MCUP (Monte Carlo Uncertainty Propagation).

>The paper now refers to topologically distinct or topologically similar models instead of geologically compatible/incompatible.

3 Motivation

The motivation of the paper cannot be justified by the need for clustering as that would place the conclusion before the premise. This paper is a proof of concept that the most basic topological signatures may be clustered into topologically coherent groups. This fact had to be observed before it is ever considered as an appropriate method of analysis.

4 Heteroscedasticity

Correct, dependency of errors is related to the type of data that is collected and should not be systematically assumed independent. Miscalibration/drift induced bias is not a form of interdependence of errors. This kind of errors stem from the instrumentation rather than from the measurements themselves. The next measurement might be biased the exact same way as the previous one yet not be dependent on it. There will be a certain level of correlation between the two that depends on the severity of the bias. Essentially, both measurements are partially dependent on the bias rather than each other. That is not to say that physical measurements are always independent, seismic data does display such properties. Assertions about heteroscedasticity being linked to dependency were removed.

5 Topology

The diagonal actually encodes the existence of a unit. It is fairly common to filter out any model with a missing unit (non-unit diagonal) although not mandatory. Therefore,

the diagonal should be retained by default. The number of units in a model may vary from the original model number down to a single unit.

6 Cluster Entropy

The equation is not only incorrect, it is irrelevant to this application. The correct equation was put in place.

7 Minor remarks

>A model can be abstracted to its contact surfaces of triple lines in an attempt to decrease its dimensionality while retaining most of its information.

>Removed.

>Figure 7 includes such table. Other captions updated for clarity.

>Yes and no. This process tends to generate a long trail of unit clusters with very low statistical significance. However, the uncertainty analysis is performed on a per cluster basis and is therefore unaffected by their removal. That is, adding or removing a cluster does not affect the results of the remaining ones.

>Unfortunately, we did not have access to this material at the time of submission nor do we have access at this date.

>References to the appendices were added inline.

8 References

Relevant references were added.

9 Others

See the marked-up version of the manuscript for more details.

---

## Author Comment (AC2) · 13 Aug 2019

1 General comments

>The MCUE procedure does not inherently assume data inputs to be independent. This was merely a working hypothesis for this particular case study. Any level of dependency may be added to the perturbation via hypersampling. A round of independent perturbation is first applied to all data and is then followed by the application of a bias/drift function to the data. The bias is itself drawn from a standard distribution of the same type as that one used in the first round. Manuscript updated for clarity.

>That is most definitely correct. Although, it does not constitute a limitation of the method per se. Rather it is a limitation of the GeoModeller API in its current itera-

tion. More complete topological signatures could not be extracted easily. This point is actually already mentioned in the discussion.

2 Detailed comments

2.1 Basic concepts

>Both acronyms removed.

>Combining all models into a single PGM or UIM implies an assumption of population singleness.

2.2 Source of uncertainties

>Disturbance distributions may be parameterized freely to account for any type of quantifiable uncertain tie. Other types of uncertainties such as conceptual uncertainty or technical failure are not statistical and are therefore out of scope of this paper.

>This point was brought by G. Caumon too. As mentioned in a previous comment, MCUE may be parameterized freely and independence was actually a practical assumption. This has been clarified in the manuscript.

>Measurements values may indeed be fully or partially dependent when taken on the same structure. Although, this does not mean that errors have to be. Both concepts are separate. For example, seismic horizon picks over a lithological interface would be measurement and error dependent. Conversely, regular compass measurement over the same structure would be measurement dependent only. The assumption is now stated clearly.

2.3 Unimodality

>The computation itself is modality neutral. However, expressing the dispersion of a dispersion with a single scalar entails that multimodal distributions will become ambiguous. Therefore, the usage of both indicators as a proxy for uncertainty, in fact, assumes unimodality.

>See previous comment.

>Homogeneity is here defined as population singleness. Manuscript was updated for clarity.

2.4 Linearity

>The perturbation process induces changes in the vector field function from which a single iso-potential surface is extracted to represent the top or bottom of the modelled formation. These changes are linear because the co-kriging algorithm used to interpolate the field is also linear. This effect is not related to the choice of disturbance distribution. The geometrical ruleset is what introduces piecewise non-linearity, not the perturbation process itself. This non-linearity is present in any implicit model which comprises a fault and/or more than one formation. Altering the geometrical ruleset is of course very likely to produce non-linear changes, however this process is outside the scope of the paper.

>Removed.

2.5 Topological signature

>Discretization is a constraint rather than an active design decision in this instance. The GeoModeller API does not provide any function point to extract topological information directly.

>Unfortunately, the GeoModeller API did not offer a practical way to extract this information at the time. However, one can assume that the discriminatory power of topology would be improved by including faulted contacts/intersections. On this basis, the proposed method would also improve.

>Correct, the relevance of the unit diagonal is now mentioned when the concept of topological signatures is introduced.

>Removed.

2.6 Entropy

>Equation was incorrect, now fixed.

2.7 Post-process

>The method described is actually being proposed rather than demonstrated. The project ran out of funds and time to complete this part. It can be removed without compromising the paper if necessary.

>The concept of thresholding is based on the premise that the method should attempt to guarantee statistical significance of the uncertainty models for each cluster. Small sized clusters, provided that stationarity is verified, are statistical outliers and do not provide any insight in terms of uncertainty. The exact threshold value of 60 was, however, a matter of convenience.

2.8 References

>We have been using the official Copernicus EndNote template to format the references. https://publications.copernicus.org/for_authors/manuscript_preparation.html. We expect the editing team to handle this matter.

>See previous comment.

>Fixed.

>The 3 appendices are now appropriately called inline.

2.9 Grammar

>Rewritten.

>Removed.

>Groundtruthing is an accepted term used in Geophysics and GIS communities. As for its formality in the common language, a reference can also be found in the Oxford dictionary. All subsequent minor grammar issues were addressed in the manuscript.
2.10 Image quality

>Low quality images were replaced with suitable ones.

>Idem.

2.11 Figure 3

>Figure 3 was removed.

>See previous.

>Idem.

---

## Author Comment (AC3) · 13 Aug 2019

[revised manuscript text omitted]

**Figure 4 Procedure for topological signature extraction**

[Figure]

**Figure 5 Topological stationarity graph with example cases**

[Figure]

**Figure 6 Density-Based Spatial Clustering of Applications (DBCAN) workflow**

[Figure]

**Figure 7 CarloTopo 3D geological model with original input foliations (disks) and interfaces (points), geometrical rulesets for units and faults, and adjacency matrix. The model box is kilometric and all data is on the x=500m vertical cross-section**

[Figure]

**Figure 8 Original CarloTopo vertical cross-sections at x=250m, 500m and 750m with potential ore deposit traps or channels circled**

[Figure]

**Figure 9 Global (top row) and top 5 most significant topological signatures vertical cross-sections of information Entropy uncertainty index models  for the low input data confidence run**

[Figure]

**Figure 10 Topological stationarity graph for the CarloTopo high input data confidence run. 1:1 graph in background as reference**

[Figure]

**Figure 11 Unique topologies occurrences for the high input data confidence run with significance threshold of 60. Note that in this instance, the clustering algorithm returned every topological signature as a distinct cluster**

[Figure]

**Figure 12 Vertical cross-sections of example plausible models for the top 5 most significant topological signatures in the high input data confidence run. Major topological changes are circled**

[Figure]

**Figure 13 Global (top row) and per-cluster vertical cross-sections of information Entropy uncertainty index models  for the low input data confidence run**

[Figure]

**Figure 14 Topological stationarity graph for the CarloTopo low input data confidence run. 1:1 graph in background as reference**

[Figure]

**Figure 15 Vertical cross-sections of example plausible models for each cluster in the low input data confidence run. Major topological changes are circled**

**Table 1 Original input structural data description for the CarloTopo 3D geological model**

| | #Foliations | #Interface points |
|---|---|---|
| UpperBasin | 1 | 4 |
| LowerBasin | 1 | 4 |
| MaficCover | 1 | 7 |
| Intrusion | 8 | 11 |
| UpperMetaFold | 1 | 2 |
| MiddleMetaFold | 6 | 7 |
| LowerMetaFold | 5 | 6 |
| Basement | NA | NA |
| Fault1 | 1 | 3 |
| Fault2 | 1 | 2 |
| Total | 25 | 46 |

**Table 2 Summary of all MCUP parameters used in this study**

[revised manuscript text omitted]